# Effects of Dietary Supplementation with Dihydromyricetin on Hindgut Microbiota and Metabolite Profiles in Dairy Cows

**DOI:** 10.3390/microorganisms14010020

**Published:** 2025-12-21

**Authors:** Jie Yu, Yingnan Ao, Hongbo Chen, Chenhui Liu, Tinxian Deng, Dingfa Wang, Min Xiang, Pingmin Wan, Lei Cheng

**Affiliations:** 1Institute of Animal Science and Veterinary Medicine, Wuhan Academy of Agricultural Sciences, Wuhan 430208, China; yujiehzau@163.com (J.Y.); ayn15849077718@163.com (Y.A.); liuchenhui@wuhanagri.com (C.L.); wangdingfa@wuhanagri.com (D.W.); xiangmin@wuhanagri.com (M.X.); wanpingmin@wuhanagri.com (P.W.); 2Hubei Provincial Center of Technology Innovation for Domestic Animal Breeding, School of Animal Science and Nutritional Engineering, Wuhan Polytechnic University, Wuhan 430023, China; chenhongbo@whpu.edu.cn (H.C.); txdeng2024@whpu.edu.cn (T.D.)

**Keywords:** dihydromyricetin, dairy cows, hindgut microbiota, metabolome, metabolic homeostasis

## Abstract

High-yielding dairy cows suffer from a high metabolic load and oxidative stress, which lead to systemic inflammation and metabolic disorders, increasing the susceptibility of these cows to various production diseases. Dihydromyricetin (DMY) has demonstrated potent antioxidant and anti-inflammatory physiological functions; however, research into its application in ruminants remains limited. This study investigated whether DMY supplementation is associated with the maintenance of metabolic homeostasis through the regulation of gut microbiota and metabolite profiles. A total of 14 mid-lactation Holstein dairy cows were randomly divided into two groups (*n* = 7 per group) and supplemented with DMY at 0 or 0.05% in their basal diet for 60 consecutive days. The effects of DMY on the blood biochemical indicators and the antioxidant capacity of the dairy cows were then determined. Alterations to the gut microbiome and the fecal and plasma metabolome were analyzed through 16S rDNA sequencing and untargeted metabolomics. The results showed that DMY significantly improved the activity of serum glutathione peroxidase (GSH-Px) and presented a trend of increasing the total antioxidant capacity (T-AOC). The abundance of multiple fiber-degrading and beneficial commensal bacteria in the gut, including *Fibrobacter_succinogenes*, *Ruminococcus_albus*, and *Turicibacter*, was significantly elevated by the DMY intervention, accompanied by the upregulation of 8,11,14-eicosatrienoic acid, myricetin, dihydro-3-coumaric acid, PGE1, L-leucine, nicotinuric acid, pantothenic acid, and pyruvate in the feces and plasma. Moreover, DMY supplementation notably reduced the abundance of potential pathogenic microbes, such as Chloroflexi, Deltaproteobacteria, *RFP12*, and *Succinivibrio*, and downregulated the levels of 12-hydroxydodecanoic acid, 12,13-DHOME (12,13-dihydroxy-9Z-octadecenoic acid), 16-hydroxyhexadecanoic acid, niacin, and glycerol 3-phosphate. These differential metabolites were principally enriched in the mTOR signaling pathway; pantothenate, nicotinate, and thiamine metabolism; glutathione metabolism; and glycolysis/gluconeogenesis. In summary, dietary supplementation with DMY increased the abundance of intestinal fiber-degrading bacteria and multiple metabolites with known anti-inflammatory and antioxidant properties in the feces and plasma, and was associated with alterations in metabolic pathways involving B-vitamins, amino acids, and glutathione. This suggests a potential role for DMY in supporting metabolic homeostasis in dairy cows.

## 1. Introduction

The long-term intensive genetic selection for milk yield in dairy cows and high-concentrate feeding regimens have substantially heightened the metabolic stress associated with milk synthesis and secretion, resulting in an increased susceptibility of these cows to various production-related diseases as a consequence of the continuous improvement in milk yield [1,2]. It is generally acknowledged that oxidative stress and chronic inflammation are common issues in high-yielding dairy cows during the transition period; they are closely related to a high metabolic load, which leads to metabolic disorders in dairy cows, thereby causing a reduced production performance and economic losses [3,4]. The application of nutritional regulation strategies is considered to be potentially favorable for alleviating metabolic stress, as well as maintaining metabolic homeostasis and the health status of dairy cows [5]. Due to the drawbacks of antibiotics, such as increased bacterial resistance, drug residues, and disordered gut microbiota, phytogenic supplements as potential antibiotic alternatives have garnered substantial attention for improving the performance of animals. Flavonoids, a kind of phytochemical, have been proven to play a beneficial role in improving the growth performance and reproductivity of animals, the quality of livestock products, and inflammatory diseases through their antioxidant and anti-inflammatory properties as well as their modulation of immune function and gut microflora [6,7,8]. In particular, the gastrointestinal microbiota functions crucially in mediating the beneficial effects of multiple plant flavonoids on host health, including citrus flavonoids, alfalfa flavonoids, and moringa leaf flavonoids [9,10,11,12,13].

Dihydromyricetin (DMY) is a flavonoid monomeric compound with multiple phenolic hydroxyl groups; it accounts for almost 30–40% (*w*/*w*) of the stems and leaves of *Ampelopsis grossedentata*, a traditional Chinese herb also called vine tea that is widely used as a medicine and health supplement [14,15]. However, DMY is not a native component of conventional ruminant feedstuffs, such as corn silage, alfalfa hay, or cereal-based concentrates. Previous studies have shown that DMY not only possesses the anti-inflammatory and anti-oxidative activities of flavonoids, but it also exerts a variety of physiological functions, such as anti-tumor functions, the regulation of glucose and lipid metabolism, and the protection of liver function [16,17,18,19], making it a promising dietary intervention for enhancing metabolic health in lactating dairy cows. The results of Wang et al. [20] demonstrated that DMY strengthened the antioxidant capacity of bovine mammary epithelial cells (BMECs) by increasing the activities of superoxide dismutase (SOD) and catalase (CAT), thereby suppressing heat-stress-induced ROS generation and oxidative stress. Dietary supplementation with DMY improved lipid metabolism and significantly enhanced SOD and GSH-Px activities in the serum and liver of mice [21]. Moreover, Fan et al. [22] systematically investigated the interaction between DMY and gut microbiota through a metabolism experiment in vitro, and confirmed that DMY could be further metabolized by fecal microflora via reduction and dehydroxylation pathways, suggesting that the gut microbiota plays an essential role in the pharmacokinetics of DMY. In an in vivo efficacy study, an extract of *Ampelopsis grossedentata* inhibited intestinal epithelial barrier injury and attenuated CCl_4_-induced liver injury by enhancing *Ruminococcaceae_UCG-014* and *Eubacterium_fissicatena_group*, as well as by decreasing the contents of *Helicobacter* and *Oscillibacter* in the gut of mice [23]. In addition, *Ampelopsis grossedentata* extract fails to alleviate liver injury in mice with gut microbiota depletion using antibiotic treatment [23]. It is crucial to note that these findings are derived primarily from monogastric models or in vitro study. The pharmacokinetic data and direct evidence for the effects of DMY in ruminants, particularly dairy cows, are currently scarce. Due to the lack of significant amounts of DMY in conventional ruminant feedstuffs, and the absence of established dosing regimens for DMY in ruminants, it is necessary to supplement this flavonoid with an appropriate dosage in the diet and evaluate its potential benefits for dairy cows.

It is generally accepted that ruminal microbiota plays a significant role in the metabolism of various flavonoids through reactions such as reduction and dehydroxylation, etc. [24], which limits their bioavailability. However, previous reports in rats showed that DMY partially diffused into blood after being absorbed and metabolized within the gastrointestinal tract [25]. With regard to ruminants, the rumen-unmetabolized prototypical compound or its bioactive metabolites that reach the hindgut possess the potential to influence gut microbiota and metabolite profiles. Therefore, based on previously reported bioactivities of DMY in other models, we sought to investigate whether DMY intervention could influence gut microbiota and systemic metabolism in lactating cows, a question previously rarely explored. The effects of dietary supplementation with DMY on serum biochemical indices, antioxidant capacity, gut microbiome, as well as fecal and plasma metabolite profiles of lactating cows were analyzed in the current study, which might provide theoretical reference for the potential application of DMY as a feed additive in dairy cows in the future.

## 2. Materials and Methods

### 2.1. Experimental Design and Sample Collection

This study was performed in the dairy farm of Institute of Animal Science and Veterinary Medicine, Wuhan Academy of Agricultural Sciences. A total of 14 lactating Holstein cows with similar milk yield (20.37 ± 4.25 kg/d, mean ± standard deviation), parity (2.2) and lactation days (149 ± 68 d, mean ± standard deviation) were randomly allotted to two groups (7 cows per group) based on these three criteria to ensure baseline homogeneity and control for key covariates, comprising control group (designated as CON) and dihydromyricetin group (designated as DMY). Before the commencement of the trial, a statistical comparison analysis confirmed that the two experimental groups (CON and DMY) showed no significant differences in milk yield, parity, and lactation days (*p* > 0.05, statistical analysis detailed in Section 2.5, Appendix A). These two groups were fed with identical basal diet supplemented with 0, 0.05% DMY, respectively. The cows were kept in separate tie stalls, allowing them free access to both feed and water. Table 1 displays the makeup of basal diet. The DMY (98% purity) utilized in this research was extracted from *Ampelopsis grossedentata*, supplied by Hubei Jinrui Biotechnology Co. LTD (Enshi Tujia and Miao Autonomous Prefecture, China). Prior to the animal trial, a study simulating rumen fermentation in vitro was conducted to determine a suitable dietary dose range for the supplementation of DMY. The dried total mixed ration, which was identical to the basal diet used in subsequent animal study, was added with DMY at 0, 0.05%, 0.1%, 0.15%, 0.2%, or 0.25% of dry matter (DM) (0.22 g DM/bottle). The bottles were incubated at 39 °C for 12 h, 24 h, and 48 h. Fermentation parameters, including the production of gas, pH levels, the degradation rate of DM, and the concentrations of ammonia nitrogen (NH3-N) and short-chain fatty acids (SCFAs), were measured. The results showed that higher doses of DMY (0.2%, 0.25%) presented a dose-dependent inhibitory effect on key fermentation parameters, including a significant decrease in SCFAs (acetic acid, propanoic acid, butyric acid) and NH3-N, which suggested a more positive impact on in vitro rumen fermentation at the addition level of 0.05%, 0.1%, 0.15% DMY. Based on the results of in vitro study, and considering the lack of prior pharmacokinetic data in ruminants, as well as the conservative safety considerations for the first application of DMY in lactating dairy cows, DMY was supplemented at 0.05% in the basal diet for cows in DMY group. The duration of the feeding trial was 60 days in a row.

At the end of the trial (day 60), blood and fecal samples were collected from all the experimental cows. Around 10 mL of blood was drawn from the tail vein of cows into vacutainer with or without heparin sodium for subsequent separation of plasma or serum samples. In brief, for the separation of serum, the blood samples were kept and coagulated on ice and then centrifuged at 3000× *g* for 10 min at 4 °C using a refrigerated centrifuge (Hunan Xiangyi Laboratory Instrument Development Co., Ltd., Changsha, China). The supernatants were collected and aliquoted into 1.5 mL centrifuge tubes for storage at −20 °C; for plasma separation, the blood samples were inverted and mixed thoroughly, then kept on ice for 1 h followed by centrifugating at 3000× *g* for 10 min at 4 °C. The obtained supernatants were aliquoted into 2 mL cryotubes and snap-frozen in liquid nitrogen. Sterile long-sleeved gloves were used to collect fresh fecal samples directly from the rectum, serving as a representation of the hindgut microbial community, which has been widely used as an non-invasive sampling method and well documented in research related to the gut microbiome [13,26,27]. Each fecal sample was immediately aliquoted into three sterile 5 mL cryovials. All collected fecal samples were snap-frozen in liquid nitrogen and stored at −80 °C until analysis.

### 2.2. Serum Biochemical and Antioxidant Indices Measurements

Serum concentrations of total protein (TP), albumin (ALB), globulin (GLB), high-density lipoprotein cholesterol (HDL-C), low-density lipoprotein cholesterol content (LDL-C), triglyceride (TG), total cholesterol (TC) and uric acid (UA) were measured by employing a fully automatic biochemical analyzer (TBA120FR, Canon, Japan). Total antioxidant capacity (T-AOC) (Cat#A015-2), superoxide dismutase (SOD) (Cat#A001-3), glutathione peroxidase (GSH-Px) (Cat#A005-1), catalase (CAT) (Cat#A007-1), malondialdehyde (MDA) (Cat#A003-1), alkline phosphatase (AKP) (Cat#A059-2), non-esterified fatty acid (NEFA) (Cat# A042-2) and lactate dehydrogenase (LDH) (Cat#A020-2) activities/concentrations were determined using specific commercially available kits (Nanjing Jiancheng Bioengineering Institute, Nanjing, China) according to the protocols of manufacturer.

### 2.3. Fecal Microbial DNA Extraction, 16S rDNA Amplicon Sequencing, and Bioinformatic Analysis

Total genomic DNA was extracted from fecal samples using the MagBeads FastDNA Kit (116570384, MP Biomedicals, CA, USA) according to the manufacturer’s protocol. The assessment of DNA quality was conducted through 1.0% agarose gel electrophoresis, followed by quantification using a NanoDrop NC2000 spectrophotometer (Thermo Fisher Scientific, Waltham, MA, USA). From the extracted DNA, V3-V4 hypervariable region of the bacterial 16S rRNA gene was amplified using 338F and 806R primers. Sample-specific 7-bp barcodes were incorporated into the primers for multiplex sequencing. Vazyme VAHTSTM DNA Clean Beads (Vazyme, Nanjing, China) were used to purify the resulting PCR amplicons and then quantified with the Quant-iT PicoGreen dsDNA Assay Kit (Invitrogen, Carlsbad, CA, USA). Equimolar amounts of purified amplicons were combined and subjected to paired-end 2 × 250 bp sequencing on an Illumina NovaSeq platform.

Bioinformatic analysis was performed using QIIME2 (version 2019.4) with slight modification following the official tutorials. Use the cutadapt plugin of QIIME2 to remove the primer fragments from the paired-end sequences, discarding sequences that do not match with the primers; then, call DADA2 plugin through qiime dada2 denoise-paired for quality control, denoising, merging, and chimera removal, generating amplicon sequence variants (ASVs). The parameters for read quality filtering were as follows: minimum read length of 226 bp, Q-score cutoff for filtering = 2. Chimera removal parameters were set as follows: filter sequences with a maximum expected error exceeding 2 for the forward reads and a maximum expected error exceeding 4 for the reverse reads. Non-singleton ASVs were aligned using MAFFT, and a phylogenetic tree was framed using FastTree2. The taxonomic classification of ASVs was assigned using the classify-sklearn naive Bayes classifier within the feature-classifier plugin against the Greengenes 13.8 reference database. Alpha diversity indices (observed species, Shannon, Chao1, Simpson, Good’s coverage and Pielou’s evenness) were calculated using Mothur (version 1.31.2). The beta diversity, assessed using Bray–Curtis metrics, was visualized via principal coordinate analysis (PCoA). Permutational multivariate analysis of variance (PERMANOVA) implemented in QIIME2 was utilized to test the significant differences in overall microbial community structure between groups. The signature differential taxa among groups were determined by utilizing linear discriminant analysis effect size (LEfSe), which employs non-parametric Kruskal–Wallis tests (with a stringent significance threshold of α = 0.01 to reduce false positives) followed by pairwise Wilcoxon tests. Taxa with a linear discriminant analysis (LDA) score > 2.0 were considered discriminant biomarkers. Putative functional profiles of gut microbiota were anticipated using Phylogenetic Investigation of Communities by Reconstruction of Unobserved States 2 (PICRUSt2) based on the MetaCyc database. Differences in predicted microbial functional pathways were evaluated using STAMP software (version 2.1.3).

### 2.4. Untargeted Metabolomics Analysis of Feces and Plasma

Thawed plasma (100 μL) and fecal (50 mg) samples were individually mixed with 400 μL of an ice-cold extraction solvent mixture (methanol/acetonitrile, 1:1 *v*/*v*). Following a vortexing step for 30 s, the samples underwent sonication for 10 min in a water bath maintained at 4 °C. Protein precipitation was achieved by incubating at −40 °C for 1 h, followed by centrifugal separation at 14,000× *g* for 20 min (4 °C). The resulting supernatant was then subjected to vacuum drying and reconstituted with 100 μL of acetonitrile/water (1:1, *v*/*v*). No visible sediment was observed after reconstitution. The final supernatants obtained after a second centrifugation (14,000× *g*, 15 min, 4 °C) were used for LC-MS analysis. To ensure the reliability and reproducibility of the extraction for a broad spectrum of metabolites, quality control (QC) samples were prepared by combining 10 μL aliquots from each individual fecal or plasma extract. These QC samples were inserted at regular intervals (every 5 experimental samples) throughout the analytical sequence.

Chromatographic separation was performed on a Vanquish UHPLC system (Thermo Fisher Scientific) that included an ACQUITY UPLC^®^ HSS T3 column (2.1 mm × 100 mm, 1.8 µm, Waters, Milford, MA, USA) maintained at 40 °C, to resolve a wide range of metabolites. The mobile phase consisted of (A) 0.1% (*v*/*v*) formic acid in water and (B) 0.1% (*v*/*v*) formic acid in acetonitrile, delivered at 0.30 mL/min with a 2 μL injection volume. The gradient elution program was: 0–1 min, 0% B; 1–12 min, 0–95% B; 12–13 min, 95% B; 13–13.1 min, 95–0% B; 13.1–17 min, 0% B. This chromatographic method was optimized for the simultaneous separation of a wide range of metabolites, including amino acids, organic acids, fatty acids, bile acids, and other polar to mid-polar compounds. Mass spectrometry was conducted using a Q Exactive HFX Hybrid Quadrupole-Orbitrap mass spectrometer (Thermo Fisher Scientific) with a heated ESI source operating in both positive and negative ionization modes. Data acquisition was conducted using simultaneous MS1 and MS/MS in Full MS-ddMS2 mode, utilizing data-dependent MS/MS with the specified parameters: sheath gas, 40 arb; auxiliary gas, 10 arb; spray voltage, +3.0 kV (ESI^+^), −2.80 kV (ESI^−^); a capillary temperature of 325 °C; an MS1 scan range of m/z 100–1000; an MS1 resolution of 70,000 FWHM; an MS/MS resolution of 17,500 FWHM; with 10 data-dependent scans carried out per cycle; and a normalized collision energy of 30 eV.

Raw data were obtained using Xcalibur 4.1 (Thermo Fisher Scientific) and analyzed with Progenesis QI software (Waters, Milford, MA, USA). The processing steps included baseline filtering, identification of peaks, integration, correction of retention times, and alignment. Batch effects were minimized using the LOESS signal correction method based on QC samples. Metabolites exhibiting >30% relative standard deviation (RSD) in QC samples were excluded. In the current study, for fecal QC samples, the proportions of metabolites with RSD < 30% in POS mode and NEG mode were 72.8% and 65.9%, respectively; for plasma QC samples, the proportions of metabolites with RSD < 30% in POS mode and NEG mode were 68.5% and 70.5%, respectively. Metabolite identification was achieved by matching accurate m/z value (<10 ppm) and MS/MS spectra against public databases (HMDB, MassBank, LipidMaps, mzCloud, KEGG) and an in-house library of authentic standards. Dimension reduction analysis (e.g., PLS-DA, OPLS-DA) was executed using “Ropls” in R package (v1.22.0). The identification of differential metabolites was screened based on variable importance in projection (VIP) > 1 (derived from OPLS-DA), *p* < 0.05, and fold change > 1.2 or <0.83. For the clustering of differential metabolites, the “Pheatmap” package (version 1.0.12) was employed. KEGG pathway enrichment analysis was implemented with the “clusterProfiler” package (version 4.6.0).

### 2.5. Statistical Analysis

Baseline characteristics analysis for experimental cows, and the calculation of sample size, was conducted using SPSS 27.0 (IBM, Chicago, IL, USA). The post hoc power analysis revealed that the sample size used in this study delivered a power of 1.0 at α = 0.05 for a two-sided test. Data are expressed as mean ± standard error of the mean (SEM). The statistical analysis was conducted using GraphPad Prism 9.0 (GraphPad Software Inc., San Diego, CA, USA). Normality of data distribution was assessed using the Shapiro–Wilk test. For parametric data, differences between two groups for serum biochemical/antioxidant indices and alpha diversity indices were assessed via unpaired two-tailed Student’s *t*-test, while non-parametric data for the relative abundance of bacterial taxa were analyzed via Mann–Whitney U test. An exploratory Spearman rank correlation coefficient and significance test analysis (without multiple testing correction) between differential metabolites (feces and plasma) and fecal microbiota was performed using the correlation function in R, with results visualized via heatmap. Statistical significance was defined as *p* < 0.05, while 0.05 < *p* < 0.10 indicated a trend toward significance.

## 3. Results

### 3.1. Effects of DMY on Serum Biochemical Indices of Dairy Cows

Compared with the CON group, the content of UA was significantly increased in the DMY group (*p* < 0.05), while AKP exhibited a trend towards increase (*p* = 0.075). However, no significant differences in the concentrations of TP, ALB, GLB, ALT, AST, BUN, TG, TC, HDL-C, LDL-C, NEFA, and LDH were observed between the two groups (*p* > 0.05) (Table 2).

### 3.2. Effects of DMY on Serum Antioxidant Capacity of Dairy Cows

The results of blood antioxidant capacity analysis are presented in Table 3. Compared with CON group, cows in the DMY group exhibited significantly higher GSH-Px activity (*p* < 0.05) and a tendency toward elevated T-AOC (*p* = 0.088). Additionally, CAT and SOD activities, along with MDA content, did not differ significantly between groups (*p* > 0.05).

### 3.3. Variations in Diversity and Taxonomic Composition of Gut Microbiota Between Groups

A total of 684,072 and 700,676 raw reads were generated from the stool samples of the CON and DMY groups through 16S rDNA sequencing, respectively. Following quality filtering, denoising, merging, and chimera removal, 369,608 and 396,000 high-quality sequences were retained for the CON and DMY groups, respectively, yielding an average of 54,686 high-quality sequences per sample. Clustering of sequences at a 97% similarity threshold resulted in 20,972 non-singleton ASVs across all samples. All these ASVs were taxonomically classified into 23 phyla, 46 classes, 89 orders, 153 families, 282 genera, and 353 species. The rarefaction curves for all samples reached a plateau phase, demonstrating sufficient sequencing depth and comprehensive coverage of the fecal microbiota in the current study (Appendix A). Alpha diversity metrics, which estimate the richness, uniformity and diversity of microflora, indicated no significant differences in observed species, Good’s coverage, Pielou_e, Chao1, Shannon, and Simpson indices between the CON and DMY groups (*p* > 0.05) (Appendix A). The beta diversity of fecal microbiota, assessed by performing principal coordinate analysis (PCoA) based on Bray–Curtis dissimilarity, showed no obvious separation between the CON and DMY groups (Appendix A). Permutational multivariate analysis of variance (PERMANOVA) revealed a trend toward reduced microbial community dispersion in the DMY group compared to the CON group (*p* = 0.075), suggesting a more homogeneous fecal microbiota among dairy cows fed with DMY (Appendix A).

Taxonomic composition analysis revealed that, at the phylum level, Firmicutes (64.54%, 62.39%), Bacteroidetes (30.66%, 32.67%), Tenericutes (1.14%, 1.70%), Spirochaetes (0.89%, 1.12%), and Actinobacteria (0.67%, 0.62%) were predominant in the gut of cows in the CON group and DMY groups, respectively (Figure 1a; Appendix A). At the genus level, *5-7N15* (3.20%, 2.72%), *CF231* (2.75%, 2.96%), *Oscillospira* (2.65%, 2.33%), *Clostridium* (1.98%, 2.19%), *Ruminococcus* (1.91%, 1.99%), *Treponema* (0.72%, 0.94%), *Dorea* (0.67%, 0.97%), *Butyrivibrio* (0.87%, 0.69%), etc., constituted the most abundant microbiota in the CON and DMY groups, respectively (Figure 1b; Appendix A).

### 3.4. Analysis of Gut Differential Abundant Taxa and Microbial Function Potential of Dairy Cows in Different Groups

To characterize the specific differential microbial taxa altered by DMY supplementation, LEfSe analysis and LDA were performed on the fecal microbiota profiles. A cladogram illustrating the phylogenetic distribution of differential taxa is shown in Figure 2a, while a histogram of LDA scores (LDA > 2.0) highlighting discriminant features between the CON and DMY groups is presented in Appendix A. Compared with the CON group, the relative abundance of Chloroflexi (*p* < 0.05), Deltaproteobacteria (*p* < 0.05), *Succinivibrio* (*p* < 0.05) was significantly decreased in the DMY group. In contrast, the relative abundance of Fibrobacteres (*p* < 0.05), *Turicibacter* (*p* < 0.05), *Corynebacterium_simulans* (*p* < 0.05), *Fibrobacter_succinogenes* (*p* < 0.05) and *Ruminococcus_albus* (*p* < 0.05) was notably increased in the gut of cows fed with DMY (Figure 2b).

Putative PICRUSt analysis was employed to investigate DMY-induced potential functional shifts in the gut microbiota of dairy cows. As depicted in Figure 2c, MetaCyc pathways related to glycolysis (*p* = 0.027) and carbohydrate biosynthesis (*p* = 0.042) were significantly enriched in the DMY group. A tendency toward enhanced functional capacity was observed for phospholipases (*p* = 0.092) and nucleoside and nucleotide biosynthesis (*p* = 0.098). Conversely, metabolic regulator biosynthesis (*p* = 0.059) and inorganic nutrient metabolism (*p* = 0.072) exhibited decreasing trends in the DMY group (Figure 2c).

### 3.5. Fecal Metabolomic Profiling of Dairy Cows in Different Groups

Untargeted metabolomic profiling was utilized for characterizing the fecal metabolic alterations in dairy cows following DMY supplementation. A total of 11,431 feature peaks were detected under both positive and negative ionization modes, resulting in the identification and annotation of 623 metabolites against reference databases. Multivariate statistical analysis demonstrated distinct clustering patterns between the CON and DMY groups in PLS-DA score plots for both positive and negative ionization modes (Appendix A), indicating significant metabolic shifts induced by DMY intervention. Volcano plot analysis identified 19 differentially abundant metabolites, including 15 upregulated and 4 downregulated metabolites (Figure 3a; Appendix A). Hierarchical clustering analysis of each sample revealed an obvious partition in the composition of fecal differential metabolites between groups (Figure 3b). These differential metabolites can be classified into several categories, comprising carboxylic acids and derivatives (10.5%), benzene and substituted derivatives (10.5%), fatty acyls (5.3%), steroids and steroid derivatives (5.3%), and organonitrogen compounds (5.3%) (Appendix A). Compared to the CON group, DMY administration significantly elevated the abundance of 8,11,14-eicosatrienoic acid (*p* < 0.001), beta-alanyl-L-arginine (*p* < 0.05), L-leucine (*p* < 0.01), cortisol (*p* < 0.05), cytosine (*p* < 0.05), and D-arabitol (*p* < 0.05), while reducing the abundance of delta-tocotrienol (*p* < 0.05) and cis-4-hydroxy-D-proline (*p* < 0.05) (Figure 3c).

Receiver operating characteristic (ROC) analysis, a commonly used tool for evaluating the performance of classification models, identified N,N-dimethylsphing-4-enine (AUC = 1), 8,11,14-eicosatrienoic acid (AUC = 0.93), L-leucine (AUC = 0.91), 4-quinolinecarboxylic acid (AUC = 0.8367) as high-discriminatory biomarkers for distinguishing differences between the CON and DMY groups (Figure 4a–d). KEGG enrichment analysis further demonstrated that the profiled fecal differential metabolites were mainly enriched in pathways, such as the mTOR signaling pathway, pyrimidine metabolism, cortisol synthesis and secretion, branched-chain amino acid biosynthesis, and linoleic acid metabolism (Figure 4e).

### 3.6. Plasma Metabolomic Profiling of Dairy Cows in Different Groups

A total of 14,735 feature peaks, including 6918 in positive ion mode and 7817 in negative ion mode, were captured in all plasma samples. After matching with the database, 6753 metabolites were confidently annotated. PLS-DA analysis of metabolite abundance across samples revealed a clear separation between the CON and DMY groups under both ionization modes (Appendix A). A volcano plot presenting the distribution of differential metabolites is shown in Figure 5a. DMY intervention resulted in alterations to 409 plasma metabolites, of which 230 were upregulated and 179 were downregulated (Figure 5a; Appendix A). Hierarchical clustering analysis demonstrated a clear divergence in the composition of differential metabolites between the CON and DMY groups, with the majority of differential metabolites exhibiting higher relative abundance in the DMY group (Figure 5b). Statistical classification results suggested that the plasma differential metabolites in the CON_vs_DMY group primarily included fatty acyl (6.6%), carboxylic acids and derivatives (4.2%), prenol lipids (2.9%), and organooxygen compounds (2.2%) (Appendix A). In particular, the key significantly altered plasma metabolites are summarized in Table 4.

The results of ROC analysis indicated that dihydro-3-coumaric acid (AUC = 0.836), prostaglandin E1 (AUC = 0.877), nicotinuric acid (AUC = 0.918), niacin (AUC = 0.877), pantothenic acid (AUC = 0.816), pyruvate (AUC = 0.877), 12,13-DHOME (AUC = 0.938), and 12-Hydroxydodecanoic acid (AUC = 0.816) were identified as typical signature differential metabolites in the plasma of dairy cows administrated with DMY (Figure 6a). KEGG enrichment analysis revealed that the plasma differential metabolites induced by DMY intervention predominantly focused on pyrimidine metabolism, pantothenate and CoA biosynthesis, thiamine metabolism, nicotinate and nicotinamide metabolism, taurine metabolism, linoleic acid metabolism, glycolysis/gluconeogenesis, and glutathione metabolism (Figure 6b).

### 3.7. Spearman Correlation Analysis Between Differential Fecal Microbiota and Metabolites in Feces

The suggestive correlations between DMY intervention-enriched gut differential microbes and fecal differential metabolites were further dissected by applying Spearman correlation coefficients and significance testing. The results indicated that Chloroflexi and Deltaproteobacteria were significantly negatively correlated with 8,11,14-eicosatrienoic acid and cortisol. Deltaproteobacteria and *Succinivibrio* exhibited significant negative correlations with L-leucine. Fibrobacteres, Tenericutes, Mollicutes, and *Fibrobacter succinogenes* demonstrated significant positive correlations with L-leucine. *Turicibacter* and *Ruminococcus albus* correlated positively with D-arabitol. *Ruminococcus albus* displayed significant positive correlations with 8,11,14-eicosatrienoic acid, D-arabitol, and cytidine (Figure 7a).

### 3.8. Associations Between Significantly Different Fecal Microbiota, Blood Indices and Metabolites in Plasma

Chloroflexi, which was significantly downregulated by DMY intervention, negatively correlated with T-AOC, GSH-Px, dihydro-3-coumaric acid, and pyruvate while positively correlated with maleic acid, ellagic acid, and niacin. *Succinivibrio* was observably negatively correlated with UA and nicotinuric acid but significantly positively associated with stearic acid, 16-hydroxyhexadecanoic acid, and 12,13-DHOME. However, *Turicibacter*, which was significantly enriched by DMY intervention, demonstrated positive correlations with GSH-Px and pyruvate, while exhibiting a significant negative correlation with 12,13-DHOME. *Ruminococcus albus* demonstrated significant negative correlations with maleic acid, ellagic acid, and stearic acid but significant positive correlations with glutamine and nicotinuric acid. *Corynebacterium simulans* presented significant positive correlations with UA, AKP, GSH-Px, and prostaglandin E1 but significant negative correlations with maleic acid, 4-methoxycinnamic acid, 16-hydroxyhexadecanoic acid, 12,13-DHOME, and niacin (Figure 7b).

## 4. Discussion

The metabolic stress associated with high milk production, coupled with the challenges posed by long-term high-concentrate feeding, exacerbates oxidative stress in dairy cows during lactation, leading to reduced milk yield and increased susceptibility to diseases [2,3,28]. Therefore, nutritional strategies, including dietary antioxidant supplementation, are critical. In this study, DMY significantly elevated serum UA levels and GSH-Px activity, with a trend towards increased AKP and T-AOC observed. UA is an endogenous metabolite that indicates the status of purine metabolism [29]. Sun et al. [30] revealed that DMY significantly upregulates UA levels and inhibits its renal reabsorption, thereby ameliorating hyperuricemia. Although DMY leads to a significant rise in serum UA here, the values remain within normal physiological ranges, suggesting DMY may influence purine recycling—a point warranting further study using specific purine metabolism markers. AKP is a well-established indicator linked to liver function and bone metabolism [31,32]. The observed trend of increased AKP activity (*p* = 0.075) in DMY-fed cows may reflect enhanced hepatic detoxification or bone turnover, processes that are highly relevant during the metabolic demands of lactation. Given previous reports on DMY’s hepatoprotective and metabolic modulatory effects [18,19], this observation implies a potential area for further investigation. The observed increase in GSH-Px and T-AOC further confirms the antioxidant capacity of DMY in ruminants, consistent with the findings of Wang et al. [20].

An increasing body of evidence underscores that flavonoids contribute to maintaining intestinal and systemic health by modulating gut microbiota [33,34]. Our study showed that DMY significantly reduced the abundance of Chloroflexi, Deltaproteobacteria, and *Succinivibrio*. Deltaproteobacteria and *Succinivibrio* belong to the Proteobacteria phylum, which is often considered a microbial signature of gut dysbiosis and contains various opportunistic pathogens [35]. Specifically, *Succinivibrio* is a core rumen taxon that is negatively correlated with milk yield [36] and has been found to be enriched in cows with mastitis linked to subacute ruminal acidosis (SARA) [37]. Thus, the enrichment of *Succinivibrio* and Deltaproteobacteria in the gut of cows may negatively affect lactation performance and intestinal health. Chloroflexi, which is predominant in deep-sea sediments [38], is less understood in the mammalian gut but harbors pathways for degrading recalcitrant compounds [39].

Additionally, Fibrobacteres, *Turicibacter*, *Fibrobacter succinogenes*, *Ruminococcus albus*, etc., were notably enriched in the gut of DMY-fed cows. *Turicibacter* plays a role in modifying host bile acids and lipid metabolism [40] and has been positively correlated with milk yield in primiparous cows [41]. *F.succinogenes* and *R. albus* are dominant ruminal cellulolytic bacteria essential for degrading dietary fiber into fermentable metabolites like short-chain fatty acids (SCFAs) and succinate [42,43,44]. Their enrichment in high-yielding cows during the periparturient period is positively correlated with milk production [45]. The enrichment of *F.succinogenes* and *R.albus* induced by DMY suggests a potential enhancement in the host’s fiber degradation capacity. It is generally accepted that the metabolites produced by these bacteria, including SCFAs and succinate, not only serve as energy substrates but are also critical for gut health and immune modulation [46]. Although we did not quantitatively measure SCFAs, PICRUSt2 analysis predicted a higher potential for the glycolysis pathway within the gut microbiota of the DMY group, which is a core process for SCFA generation. Based on the predicted shifts in microbial functional potential, we speculate that the enrichment of these fiber-degrading bacteria suggests the potential for elevated SCFA production, a hypothesis that requires direct quantification of SCFA production in future studies to confirm this potential functional shift. Therefore, DMY exerted positive influences on the gut microbiome of lactating cows by reducing the abundance of potential pathogens and increasing the prevalence of beneficial bacteria

Fecal metabolomic profiling identified several signature differential metabolites in DMY-fed cows, which are typical to clarify the impacts of DMY on metabolic changes in the gut microbiota. 8,11,14-eicosatrienoic acid, also known as Dihomo-γ-linolenic acid (DGLA), is an ω-6 PUFA with anti-inflammatory activity. It can be metabolized through the cyclooxygenase pathway (COX1, COX2) into PGE_1_, which possesses significant anti-inflammatory activity, or converted via the 15-lipoxygenase pathway (LOX) into 15-HETrE, thereby exerting anti-inflammatory, antihypertensive, and anti-neoplastic activities [47]. Thus, its increase might contribute to alleviating oxidative stress and inflammation in lactating cows. N,N-Dimethylsphing-4-enine (DMS) is an endogenous sphingolipid metabolite that regulates sphingolipid signaling and metabolic homeostasis [48,49]. It has been shown to recruit regulatory T cells to protect against ischemia-reperfusion injury [50]. 4-quinolinecarboxylic acid might be an intermediate in the tryptophan–kynurenine pathway. Gut microbiota-derived tryptophan metabolites are key immune regulators via the aryl hydrocarbon receptor (AHR) pathway [51]. The increased levels of DMS and 4-quinolinecarboxylic acid in DMY-fed cows might be implicated in intestinal immune regulation, whereas the regulatory roles of DMS and 4-quinolinecarboxylic acid in the intestinal homeostasis of dairy cows require further elucidation. The upregulation of fecal L-leucine aligns with the positive effects of citrus flavonoids on ruminal amino acid metabolism [52]. An unexpected finding was the increased fecal cortisol. Cortisol is primarily a glucocorticoid hormone associated with the stress response. As plasma cortisol levels were not measured, this elevation potentially indicates altered cortisol metabolism or excretion rather than necessarily implying heightened systemic stress.

Plasma metabolomic analysis revealed that DMY significantly increased levels of metabolites with known anti-inflammatory and antioxidative properties. Myricetin, an oxidative metabolite of DMY, alleviates oxidative stress and inflammation [53,54]. Its elevated plasma levels may originate from a synergistic interplay between direct gut microbial biotransformation and host absorption of rumen-bypassed unmetabolized DMY, followed by hepatic oxidation. Further dissection is needed to definitively identify the key microbial players and pathways responsible for DMY metabolism by employing radiolabeled tracking and metatranscriptomic profiling. Dihydro-3-coumaric acid, derived from the reduction of coumaric acid, acts as a free radical scavenger [55]. The upregulation of myricetin and dihydro-3-coumaric acid might contribute to the enhanced systemic antioxidant capacity. The increase in plasma PGE1 in DMY-fed cows corresponded with the rise in its precursor, fecal DGLA. Furthermore, 15-deoxy-Delta-12,14-PGJ2-d4 (15d-PGJ2), an anti-inflammatory cyclopentenone prostaglandin, whose downstream metabolism involves conjugation with glutathione (GSH) [56,57,58,59,60], was decreased. 15d-PGJ2 exerts anti-inflammatory and antioxidant effects through multiple mechanisms, including the inhibition of NF-κB and regulation of Nrf2-HO1 signaling [57]. Notably, the decrease in plasma 15-deoxy-Delta-12,14-PGJ2-d4 corresponded with enhanced serum GSH-Px activity, suggesting that DMY might alleviate oxidative stress partly by enhancing the glutathione metabolism pathway, as also indicated by KEGG enrichment analysis of the plasma metabolome.

High-yielding dairy cows have substantial demands for NAD^+^ and CoA to support energy production during lactation. The decrease in plasma niacin, alongside the increased levels of nicotinuric acid, implies that DMY intervention activates niacin metabolism for NAD^+^ synthesis. Pantothenic acid (vitamin B5) is a crucial precursor of CoA, playing a central role in energy and fatty acid metabolism [61,62]. During the lactation period, the demand for energy and milk fat synthesis significantly increases, with CoA supporting triglyceride synthesis through the generation of acyl-CoA [63,64]. Additionally, as a component of the pyruvate dehydrogenase complex, CoA catalyzes the conversion of pyruvate to acetyl-CoA, which enters the TCA cycle to meet the high energy demands of lactation [65]. In this experiment, the increased plasma levels of pantothenic acid and nicotinuric acid are consistent with alterations in the metabolism of CoA and NAD^+^ precursors, respectively. These pathways are fundamental to cellular energy transduction, implying a modulation in these critical metabolic nodes. Therefore, we speculate that the B vitamin metabolism is potentially involved in the regulation of metabolic homeostasis in lactating cows by DMY.

Furthermore, DMY also downregulated multiple plasma lipid metabolites. The reduction of 4-Methoxycinnamic acid and maleic acid may be associated with the downregulation of potentially harmful bacteria in the gut of cows by DMY, given their reported antibacterial properties [66,67,68]. Hydroxylated fatty acids, such as 12-hydroxydodecanoic acid, 12,13-DHOME and 16-hydroxyhexadecanoic acid, are typically produced by cytochrome P450 enzymes that catalyze PUFAs and are closely associated with metabolic disorder and inflammation [69]. In a high-fat-diet-induced metabolic syndrome model, the level of 12-hydroxydodecanoic acid was correlated with the regulation of the gut microbiota–liver axis, suggesting its indirect involvement in the regulation of glucose and lipid homeostasis [70]. 12,13-DHOME has a dual role, promoting fatty acid uptake in brown adipose tissue [71], while also facilitating the production of inflammatory macrophages [72]. The negative correlation observed between *Turicibacter* (a potentially beneficial bacterium enriched by DMY) and 12,13-DHOME might be suggestive of this potential anti-inflammatory shift. An interesting finding was the modest downregulation of metabolites typically considered beneficial, specifically 9(Z),11(E)-conjugated linoleic acid (CLA) and spermine. While CLA is known for its anti-inflammatory properties and spermine is crucial for cell growth and protection [73,74], their metabolisms are context-dependent. The upregulation of alternative anti-inflammatory and antioxidant pathways (e.g., DGLA/PGE1, myricetin, glutathione metabolism) may reduce systemic reliance on specific pools of these metabolites. Moreover, spermine synthesis consumes S-adenosylmethionine, a major methyl donor. Under the high metabolic demands of lactation, DMY-induced modulation might promote the allocation of these resources towards other high-priority processes vital for lactation and antioxidant defense (e.g., glutathione regeneration), leading to an adjusted polyamine homeostasis. Therefore, while the observed reduction in these individual “protective factors” in DMY-fed cows might be by chance and warrant attention in future studies, it likely does not indicate a detrimental effect but rather reflects a broader metabolic reconfiguration. The net outcome, as indicated by improved antioxidant capacity and beneficial microbial shifts, appears positive, reflecting metabolic network flexibility. In terms of glucose metabolism, increased plasma pyruvate and decreased glycerol 3-phosphate might imply an enhanced glucose catabolic flux. This aligns with the PICRUSt-predicted enhancement in microbial glycolytic potential and might help maintain blood sugar stability while facilitating the allocation of glucose to the mammary gland for lactose synthesis in dairy cows, which commonly experience insulin resistance [75]. Therefore, DMY intervention was associated with alterations in pathways linked to glucose and lipid metabolism. It is important to note that our assessment of metabolic status mainly relied on metabolomics rather than direct measurements of key metabolic indicators, such as insulin or IGF-1. The observed changes in the metabolome provide a rationale for future studies to include these standard metabolic indicators alongside multi-omics approaches.

Our findings suggest an association between DMY supplementation and observed changes in gut microbiota and metabolism. However, it remains unclear whether DMY is metabolized into active forms by ruminal microbes, the extent to which DMY reaches the gut or systemic circulation intact, and the physiological implications of the decrease in certain metabolites with known beneficial activities (e.g., a specific CLA isomer, spermine). The observational nature of this study may limit the establishment of direct mechanistic causality. Future in vivo pharmacokinetic analyses and controlled intervention studies are necessary to explore the ruminal degradation patterns, absorption, distribution, and metabolism of DMY in dairy cows, as well as to confirm the specific causal relationships between DMY intervention and host metabolic changes.

## 5. Conclusions

Dietary supplementation with DMY significantly affects the hindgut microbiota and metabolite profiles of dairy cows, primarily manifested as increased serum GSH-Px activity, elevated abundance of fiber-degrading bacteria, and reduced prevalence of potential pathogenic bacteria in the intestine. Additionally, DMY intervention upregulated the levels of multiple metabolites with known anti-inflammatory and antioxidant properties in feces and plasma, also correlated with the changes in metabolic pathways involving B-vitamins, amino acids, and glutathione. These observed changes are consistent with a potential role of DMY in supporting metabolic homeostasis. It is important to note that these conclusions are drawn from an exploratory study with a 60-day intervention period, which, while sufficient to demonstrate initial effects, may be inadequate to fully evaluate the sustained effects of DMY across entire lactation stages, especially without lactation performance parameters and direct metabolic and inflammatory markers. This limitation hinders a comprehensive assessment of the effects of DMY. Therefore, the specific pharmacokinetics and mechanisms of action of DMY in the gastrointestinal tract of dairy cows need further investigation. Future larger-scale and longitudinal studies, incorporating multiple sequential sampling timepoints and direct measurement of standard metabolic indicators and production performance, are essential to elucidate the dynamic effects and physiological trade-offs, thereby validating its economic value. Collectively, the present study provides preliminary evidence and a basis for future research into the potential application of DMY as a phytogenic feed supplement in dairy cows.

## Figures and Tables

**Figure 1 microorganisms-14-00020-f001:**
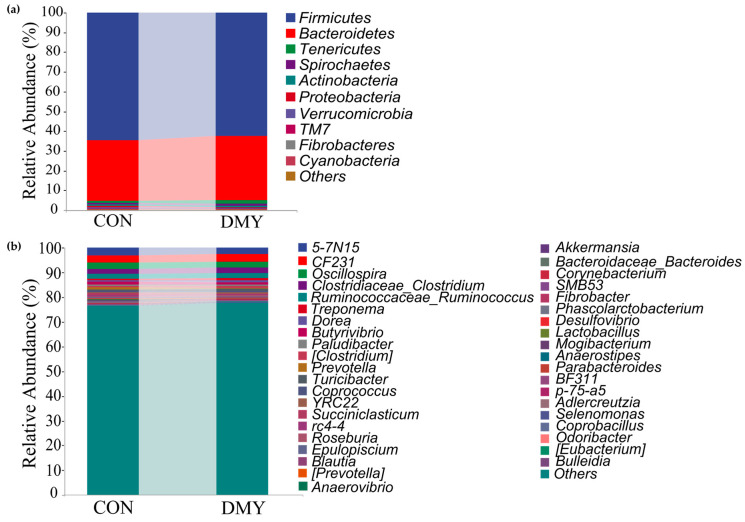
Variations in taxonomic composition of gut microbiota dairy cows fed with DMY. (**a**) Comparison of the relative abundance of top 10 phyla; (**b**) comparison of the relative abundance of top 40 genera.

**Figure 2 microorganisms-14-00020-f002:**
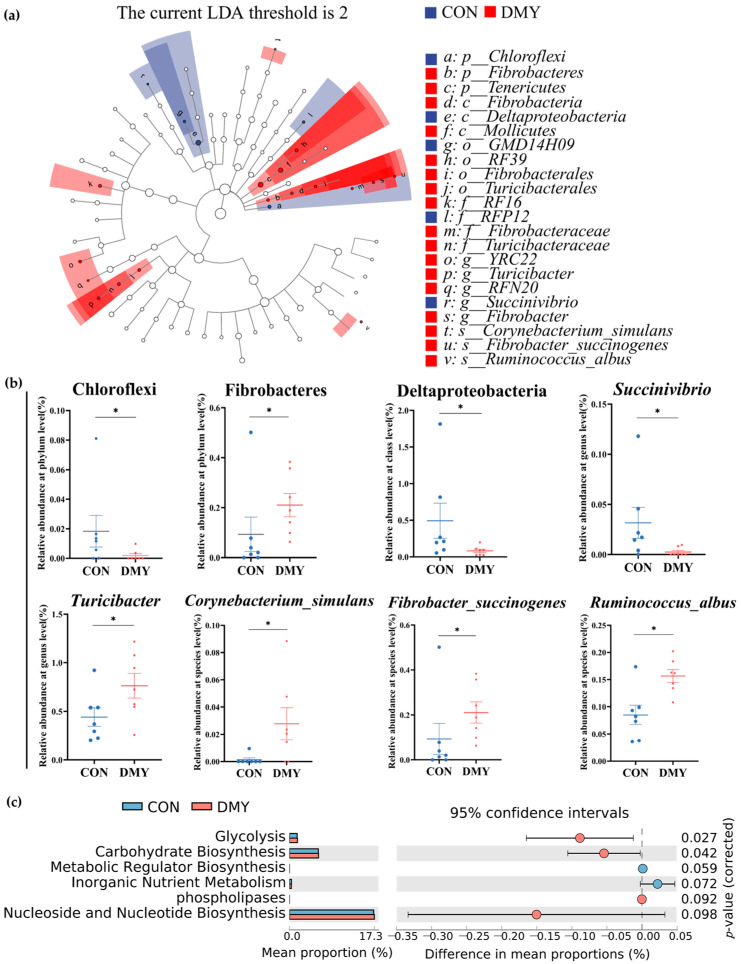
Shifts in the gut microbial composition and predicted functional potential in dairy cows induced by DMY intervention. (**a**) The cladogram of linear discriminant analysis effect size (LEfSe) of differential gut microbes between CON group and DMY group; (**b**) the changes in relative abundance of differential bacteria at several taxonomic levels. Data are presented as mean ± SEM (*n* = 7), * *p* < 0.05; (**c**) PICRUSt analysis on the difference in predicted gut microbial functional potential between CON group and DMY group.

**Figure 3 microorganisms-14-00020-f003:**
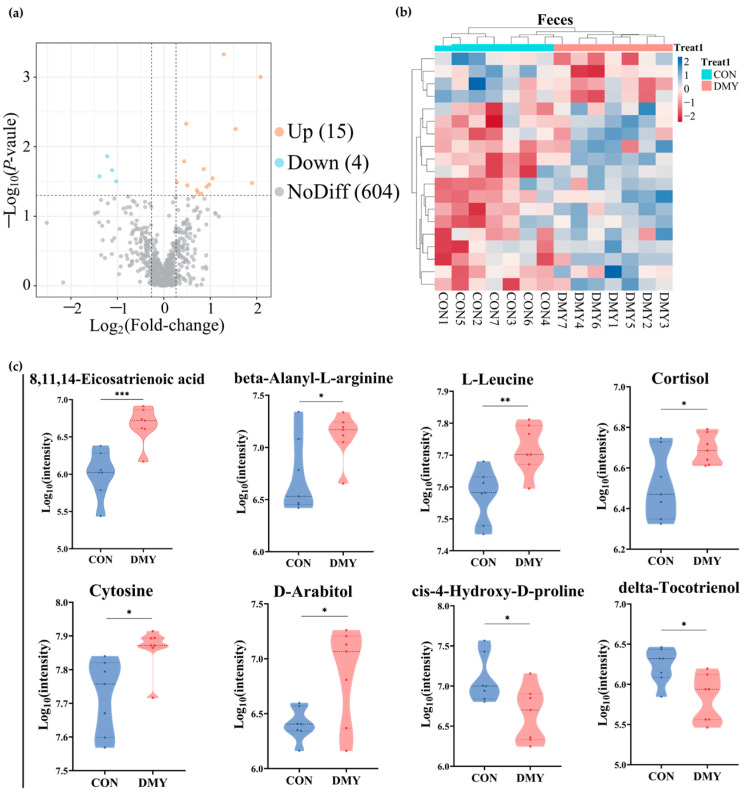
Changes in the fecal metabolome of dairy cows induced by DMY intervention. (**a**) The volcano plot of significantly differential metabolites in feces. The orange points represent significantly upregulated metabolites, the blue points indicate significantly downregulated metabolites, and the grey points signify non-significant metabolites. The horizontal dotted line represents the significance threshold (*p*-value < 0.05), and the vertical dotted line indicates the fold-change threshold (fold-change > 1.2 or <0.83); (**b**) heatmap depicting the relative abundance of all significantly differential fecal metabolites across individual samples. Rows represent differential metabolites, and columns represent individual samples, grouped by treatment. The color gradient from red to blue indicates low to high relative abundance; (**c**) comparison of the relative abundance of major differential metabolites in feces. Data are presented as mean ± SEM (*n* = 7), * *p* < 0.05, ** *p* < 0.01, *** *p* < 0.001.

**Figure 4 microorganisms-14-00020-f004:**
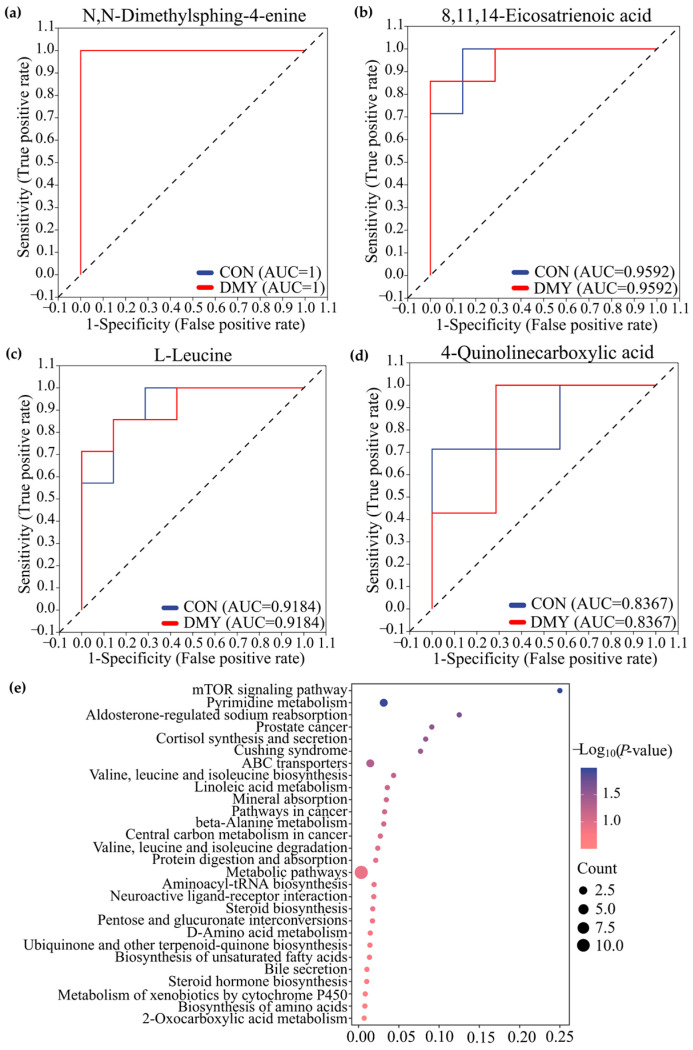
Characterization and pathway enrichment analysis of typical differential metabolites in feces. ROC curves of N,N-dimethylsphing-4-enine (**a**), 8,11,14-eicosatrienoic acid (**b**), L-leucine (**c**), 4-quinolinecarboxylic acid (**d**); (**e**) KEGG enrichment analysis for the differential metabolites in feces.

**Figure 5 microorganisms-14-00020-f005:**
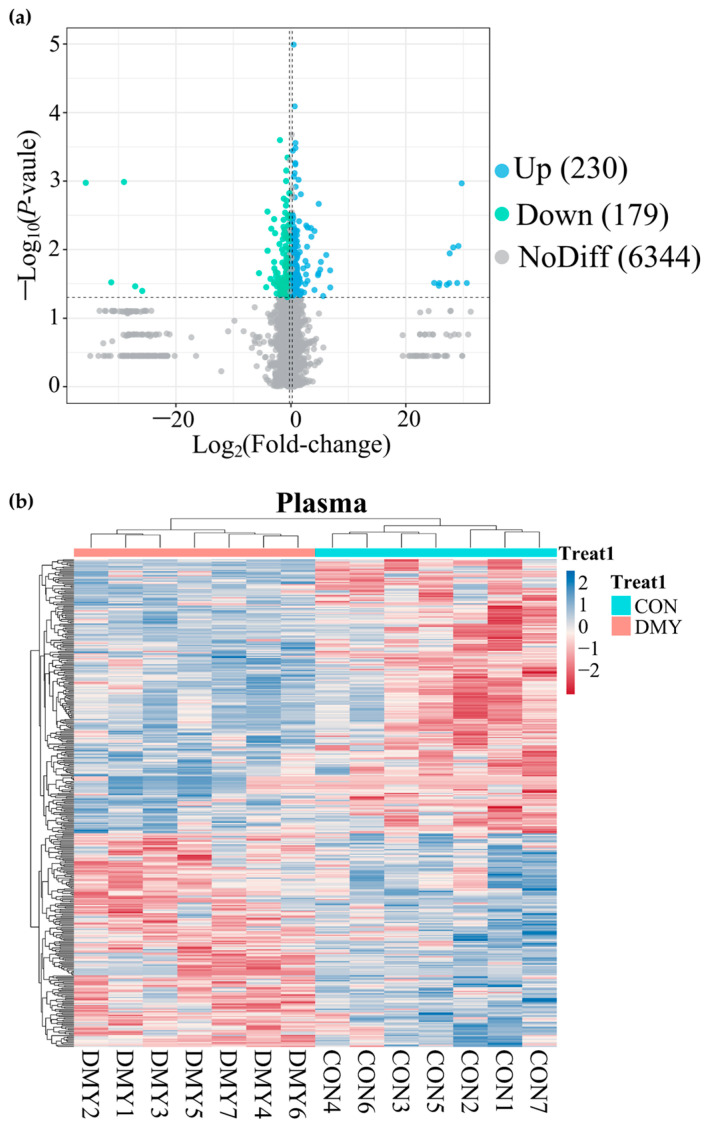
Overview of plasma metabolomic alterations in dairy cows induced by DMY intervention. (**a**) The volcano plot displaying significantly differential metabolites in plasma. The blue points represent significantly upregulated metabolites, the green points indicate significantly downregulated metabolites, the grey points signify non-significant metabolites. The horizontal dotted line represents the significance threshold (*p*-value < 0.05), and the vertical dotted line indicates the fold-change threshold (fold-change > 1.2 or <0.83); (**b**) heatmap depicting the relative abundance of all significantly differential plasma metabolites across individual samples. Rows represent differential metabolites, and columns represent individual samples, grouped by treatment. The color gradient from red to blue indicates low to high relative abundance.

**Figure 6 microorganisms-14-00020-f006:**
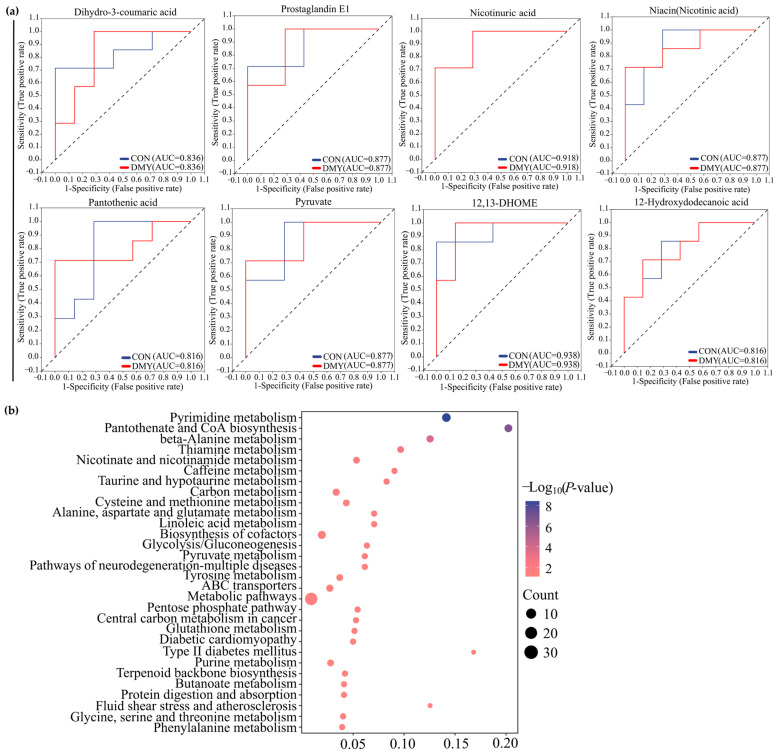
Characterization and pathway enrichment analysis of representative differential metabolites in plasma. (**a**) ROC curves of dihydro-3-coumaric acid, prostaglandin E1, nicotinuric acid, niacin, pantothenic acid, pyruvate, 12,13-DHOME, and 12-hydroxydodecanoic acid; (**b**) KEGG enrichment analysis for the differential metabolites in plasma.

**Figure 7 microorganisms-14-00020-f007:**
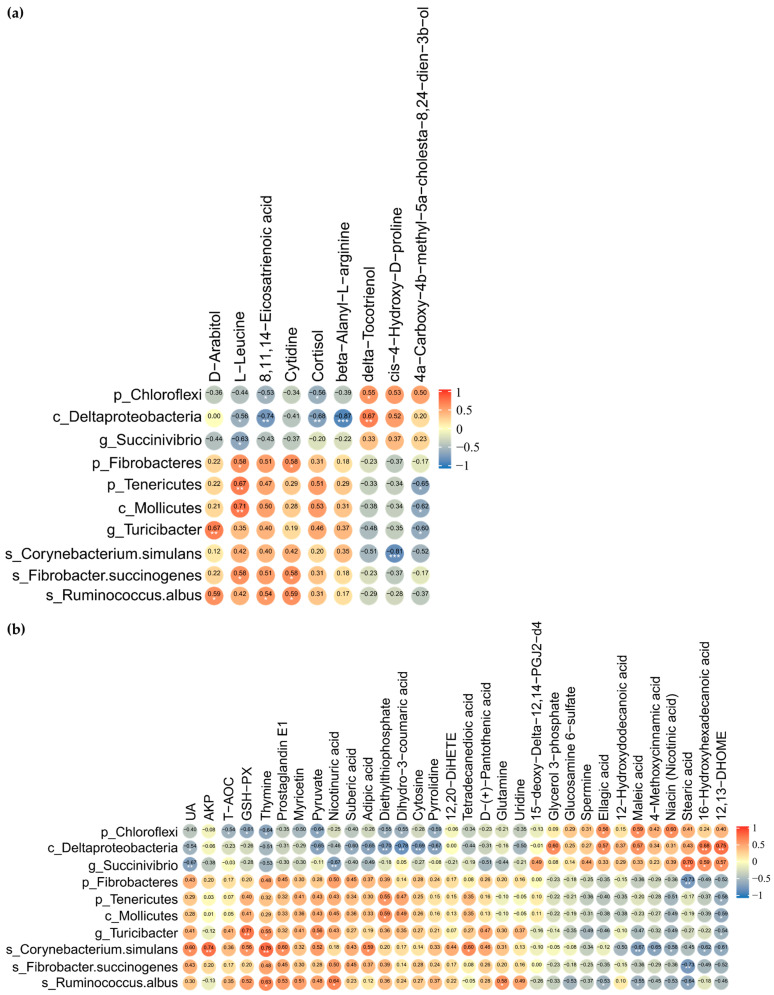
Correlation analysis of gut microbiome with fecal and plasma metabolome. (**a**) Heatmap of Spearman rank correlation coefficient and significant tests between gut differential microbes and fecal differential metabolites; (**b**) heatmap of Spearman rank correlation coefficient and significant tests between gut differential microbes, plasma differential metabolites, biochemical indicators, and antioxidant-related indices. The orange color indicates a positive correlation. The blue color denotes a negative correlation. The correlation coefficient is displayed within every cell of heatmap. Asterisks denote significance from Spearman’s correlation analysis (* *p* <0.05, ** *p* < 0.01, and *** *p* < 0.001).

**Table 1 microorganisms-14-00020-t001:** Ingredients and nutrient levels in TMR (air-dry basis, %).

Ingredients	Content/%	Nutrient Component	Nutrient Levels
Corn silage	49.35	NEL ^2^ (MJ/kg)	6.8
Peanut vine	1.95	Crude Protein/%	15.26
Beet pulp	3.25	Neutral detergent fiber/%	42.25
Brewer’s grains	19.68	Acid detergent fiber/%	27.27
Leymus chinensis	3.25	Ca/%	1.18
Concentrate supplement ^1^	22.26	TP/%	0.52
Baking soda	0.26		
Total	100		

^1^ per kg of concentrate supplement contains: Fe 350 mg, Cu 425 mg, Mn 500 mg, Zn 2100 mg, Co 25 mg, Se 13 mg, I 5 mg, Vitamin A 110,000 IU, Vitamin D 66,000 IU, Vitamin E 2700 IU. ^2^ NEL was a calculated value, while the others were measured values.

**Table 2 microorganisms-14-00020-t002:** Effects of dietary supplementation with DMY on serum biochemical indicators of dairy cows.

Items	CON	DMY	*p*-Value
TP (g/L)	73.50 ± 3.08	73.69 ± 2.30	0.962
ALB (g/L)	33.80 ± 0.70	33.27 ± 0.56	0.566
GLB (g/L)	39.70 ± 2.96	40.41 ± 2.76	0.863
AST (U/L)	89.46 ± 17.22	83.59 ± 9.25	0.769
ALT (U/L)	31.87 ± 3.08	36.03 ± 2.92	0.346
AKP (U/L)	287.66 ± 16.33	373.24 ± 40.74	0.075
BUN (mmol/L)	7.41 ± 0.78	6.54 ± 0.51	0.369
UA (µmol/L)	31.59 ± 2.42 ^b^	47.27 ± 4.69 ^a^	0.012
TG (mmol/L)	0.15 ± 0.028	0.14 ± 0.021	0.664
TC (mmol/L)	2.78 ± 0.43	2.91 ± 0.26	0.806
HDL-C (mmol/L)	1.92 ± 0.24	1.96 ± 0.12	0.875
LDL-C (mmol/L)	0.63 ± 0.15	0.71 ± 0.12	0.704
NEFA (mmol/L)	0.094 ± 0.017	0.064 ± 0.0096	0.167
LDH (U/L)	1514.46 ± 24.67	1457.85 ± 64.98	0.431

^a,b^ Values within a row with different letters differed significantly (*p* < 0.05). Data are presented as mean ± SEM. Abbreviations: TP, total protein; ALB, albumin; GLB, globulin; AST, aspartate transaminase; ALT, alanine aminotransferase; AKP, alkline phosphatase; BUN, blood urea nitrogen; UA, uric acid; TG, triglycerides; TC, total cholesterol; HDL-C, high-density lipoprotein cholesterol; LDL-C, low-density lipoprotein cholesterol; NEFA, non-esterified fatty acid; LDH, lactate dehydrogenase.

**Table 3 microorganisms-14-00020-t003:** Effects of dietary supplementation with DMY on serum antioxidant indices of dairy cows.

Items	CON	DMY	*p*-Value
T-AOC (mM)	0.75 ± 0.035	0.82 ± 0.023	0.088
SOD (U/mL)	14.59 ± 0.61	14.71 ± 0.65	0.901
CAT (U/mL)	3.69 ± 0.56	3.45 ± 0.39	0.724
GSH-Px (U/mL)	208.63 ± 20.80 ^b^	261.06 ± 9.01 ^a^	0.039
MDA (nmol/mL)	1.59 ± 0.17	1.22 ± 0.32	0.327

^a,b^ Values within a row with different letters differed significantly (*p* < 0.05). Data are presented as mean ± SEM. Abbreviations: T-AOC, total antioxidant capacity; SOD, superoxide dismutase; GSH-Px, glutathione peroxidase; CAT, catalase; MDA, malondialdehyde.

**Table 4 microorganisms-14-00020-t004:** The key significantly altered plasma metabolites in dairy cows following DMY supplementation.

Name	FC	Log_2_FC	VIP	*p*-Value	M/Z	Retention Time (min)	Class	Sub_Class	Regulation
12,13-DHOME	0.8086	−0.3064	1.9709	0.0227	295.2284	11.8467	Fatty Acyls	Fatty acids and conjugates	Down
12-Hydroxydodecanoic acid	0.6505	−0.6205	1.7368	0.0423	215.1656	10.1018	Hydroxy acids and derivatives	Medium-chain hydroxy acids and derivatives	Down
15-deoxy-Delta-12,14-PGJ2-d4	0.5534	−0.8537	1.6423	0.0420	365.2285	8.9041	Fatty Acyls	Eicosanoids	Down
16-Hydroxyhexadecanoic acid	0.8065	−0.3102	1.9296	0.0198	253.2174	6.7664	Fatty Acyls	Fatty acids and conjugates	Down
4-Methoxycinnamic acid	0.8170	−0.2916	1.9981	0.0097	177.0562	6.8574	Cinnamic acids and derivatives	Cinnamic acids	Down
5-Hydroxykynurenamine	1.5236	0.6075	1.7355	0.0467	419.1930	4.7790	Organooxygen compounds	Carbonyl compounds	Up
9(Z),11(E)-Conjugated Linoleic Acid	0.7643	−0.3879	2.2158	0.0045	279.2333	14.8305			Down
D-(+)-Pantothenic acid	1.2556	0.3283	1.7717	0.0455	220.1181	2.9693	Organooxygen compounds	Alcohols and polyols	Up
Dihydro-3-coumaric acid	1.9033	0.9285	1.8004	0.0417	165.0559	5.9823	Phenylpropanoic acids		Up
Glutamine	1.3690	0.4531	1.7432	0.0275	145.0621	1.2077	Carboxylic acids and derivatives	Amino acids, peptides, and analogues	Up
Glycerol 3-phosphate	0.4027	−1.3122	2.1493	0.0113	217.0122	1.0026	Glycerophospholipids	Glycerophosphates	Down
L-Cysteine	1.2602	0.3337	2.1073	0.0078	165.9913	14.9401			Up
Maleic acid	0.8221	−0.2827	1.9757	0.0158	115.0038	1.0594	Carboxylic acids and derivatives	Dicarboxylic acids and derivatives	Down
Myricetin	1.2439	0.3149	1.6157	0.0361	283.0250	2.0058	Flavonoids	Flavones	Up
N-Arachidonoyl Dopamine-d8	0.6678	−0.5825	2.5498	0.0005	958.7378	13.3867			Down
Nicotinuric acid	4.4799	2.1635	2.0440	0.0188	215.0234	0.8008	Carboxylic acids and derivatives	Amino acids, peptides, and analogues	Up
Prostaglandin E1	1.4751	0.5608	1.8319	0.0455	319.2270	9.4251	Fatty Acyls	Eicosanoids	Up
Pyruvate	1.8700	0.9030	2.0936	0.0080	152.0319	0.7988	Keto acids and derivatives	Alpha-keto acids and derivatives	Up
Sebacic acid	1.2952	0.3731	1.7874	0.0380	201.1134	7.1929	Fatty Acyls	Fatty acids and conjugates	Up
Spermine	0.6784	−0.5598	1.6981	0.0434	201.2075	7.1681	Organonitrogen compounds	Amines	Down
Stearic acid	0.6847	−0.5464	2.2346	0.0032	321.2206	12.8142	Fatty Acyls	Fatty acids and conjugates	Down
Uric acid	1.7361	0.7958	2.5652	0.0003	169.0357	1.0194	Imidazopyrimidines	Purines and purine derivatives	Up

## Data Availability

The original sequencing data presented in this study can be found in the NCBI SRA database at https://www.ncbi.nlm.nih.gov/bioproject/, accessed on 18 November 2025, accession number PRJNA1295706. The raw metabolomic datasets are openly available in the OMIX database at https://ngdc.cncb.ac.cn/, accessed on 18 November 2025, accession number: OMIX012142, OMIX012144.

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
