# Peer review of "Microorganisms2026, 14(1), 20;https://doi.org/10.3390/microorganisms14010020"

_microorganisms, 2025, doi:10.3390/microorganisms14010020_

Round 1

Reviewer 1 Report

Comments and Suggestions for Authors

I sincerely appreciate the opportunity to review the manuscript “Dietary Supplementation with Dihydromyricetin Beneficially Affects Hindgut Microbiota and Metabolic Characteristics in Dairy Cows”. The study addresses a topic of considerable interest, and I value the effort undertaken by the authors. However, I have identified several areas that require attention in order to strengthen the scientific robustness of the work. I believe that a substantial revision would help clarify methodological aspects and ensure that the interpretations are fully aligned with the evidence presented.

  1. Title and Abstract

The title suggests mechanistic causality (“affects”), yet the design is observational and lacks control of environmental factors; causality is not demonstrated.
The authors mention “anti-inflammatory” effects without having measured cytokines, inflammatory eicosanoids, C-reactive protein, haptoglobin, SAA, or immune markers. On what basis is an anti-inflammatory effect claimed?

  1. Introduction

There is no justification for the chosen dose: an in vitro assay is mentioned, yet the Introduction does not connect the dose with previous studies in ruminants.
The text describes studies in mice and human cells, but lacks prior evidence in cattle to justify such direct physiological extrapolations.
The extremely low bioavailability of flavonoids in ruminants is not addressed, a key point for interpreting systemic effects.
How is ruminal degradation of DMY considered, and is there evidence that it reaches the intestine or circulation intact?
Are there pharmacokinetic data in ruminants to support the hypothesis of systemic effects?

  1. Materials and Methods

3.1 Experimental design

It is not stated whether the design accounted for random blocks, differentiated days in lactation, or nutritional covariates.

The experiment relies solely on measurements at day 60, lacking temporal resolution or kinetics.

Why were no longitudinal measurements performed, considering that the microbiota is highly dynamic?

3.2 Biochemical analyses

There is no control of daily variation, photoperiod, production status, dry matter intake, etc.

Glucose, insulin, or metabolic hormones (IGF-1): all indispensable markers of “metabolic health” were not measured.

How do the authors justify metabolic conclusions without measuring standard metabolic indicators?

3.3 Microbiota – 16S sequencing

The Greengenes 13.8 database is obsolete (last updated in 2013).
This affects taxonomic accuracy, especially for rumen genera such as Ruminococcus, Fibrobacter, Turicibacter.

A total of 20,972 ASVs per sample is reported, an unusually high number that may indicate denoising errors, contamination, incorrect DADA2 parameters, or artefacts arising from low-quality or noisy reads.

Why was Greengenes used instead of SILVA 138 or GTDB?

How was it verified that the ASVs do not include contamination?

Was correction for multiple testing (FDR) applied?

3.4 PICRUSt2

PICRUSt2 predicts functions in silico; it does not measure real functional activity.
With an obsolete taxonomic reference, the inferred functional results are unreliable.

Why is “increased glycolysis” interpreted if no fermentative metabolites such as SCFAs were measured?

3.5 Faecal and plasma metabolomics

An extremely large number of metabolites is detected (>14,000), but confirmed chemical annotation includes only 623 (faeces) and 6,753 (plasma).

Metabolites are interpreted as “anti-inflammatory”, “antioxidant” or “beneficial” without measuring cytokines or immune biomarkers.

How was the identification of prostaglandin E1 validated, considering that its detection requires specific analytical methods?

  1. Results

Serum uric acid increases yet is interpreted as a sign of “metabolic improvement”; however, it may reflect oxidative stress or altered purine recycling.

Faecal cortisol is reported as “beneficial”, although in bovine physiology it typically indicates metabolic stress.

How do the authors explain the increase in faecal cortisol and its interpretation as a positive outcome?

  1. Discussion

The Discussion conflates correlations with mechanisms without supporting evidence.
Undemonstrated assumptions include:

that DMY increases SCFAs (these were not measured);

that metabolites such as DGLA are produced through the action of Ruminococcus (no evidence is provided);

that nicotinuric acid or pantothenic acid indicate “improved energy metabolism”.

Why do the authors claim that DMY improves “glucose and lipid metabolism” without measuring glucose, insulin, NEFA, β-HBA, or metabolic hormones?
How are anti-inflammatory inferences justified without data on IL-6, LPS, Hp or SAA?
What is the plausible mechanism through which DMY would reach the colon intact, considering ruminal degradation?

 Conclusions

The conclusions exceed the experimental evidence.

Respectfully,

Author Response

Thanks for your valuable comments. Please see the attachment.

Reviewer 2 Report

Comments and Suggestions for Authors

The authors have undertab´ken the attempt to investigate the micribiota and metabolic effects of the addition of dihydromyricetin to the feeds of diary cows. The topic is highly important, but there are several deficits in this manuscript

General:

Except for the Introdiúction, the description is very entiring. This literally means that it is of inferior stylistic quality, with boring long texts rather than short pregnant description of findings, attractive tables and/or figures and a focussed discussion. The figures are overloaded, uncomely and with too small fonts. Several parts of them can be transferred to the online supplement. The conclusion is misleading. The authors write about synthesis and metabolism, but there was no synthesis and metabolism investigated. Data are merely descriptive, i.e. epiphenomena. This paper can become attractive, but in its crrrrent form it's an imposition

Specific: 

Major:

L. 49: Insert a para, why  dihydromyricetin isn't sufficient in regular dairy cow feeds.

L. 112: table s1: Basal diet data belong to the main body of manuscript.

L. 196: Is there any proof that all analysed compounds resolve quantitatively in that supernatant and the reconstituted acetonitrile/water phase? There are substantial doubts concerning many lipids! Was there a sediment after reconstitution?

L. 201: Chromatographic separation of which compounds? Specify!

L. 232, Statistics:  Indicate, whether data were analyzed for normal distribution and corrected for multiple testing.

Table 1: Methods for these parameters were not described.

L. 285ff: Create a table for these data, comparing the experimental groups. Provide data with meians/IQR or means/SD.

All Figures: Enlanrge font sizes, improve colors, reduce information in any figure presented and move any information that isn't essential for understanding the discussion into the supplement. In most figures this means removal of a-c or d, only leaving the last part (d/e) at its place. In fig 2 it means removal of a+b, leaving c at its place, with improved coloring. 

L. 389: create a table rather than describing this in the text. Provide values.

Line 439ff: present data in a table.

L. 481ff: Hence, the study is extremely underpowered. Usually, a para 'Strengths and Limitations of Study' discusses this.

The discussion must be shortened by literally 50%. 5 1/2 pages is inacceptable.

L. 731: enhanced synthesis was not determeined, neither was metabolism. There are only epiphenomenological data. Change text, please. 

Minor:

L 37; define 12,13-DHOME

L. 104: Define, whether this is means and SD or something different.

L. 108: Refer to the Statistics para here! Insert a table to show data, with reference to it.

L. 130: delete >containing<. 

L. 132: provide apparatus, and g-force. r/min isn't an adequate term!

L. 133: >supernatant was< or >supernatants were<.

L. 307: replace 'Whereas' by 'In contrast'.

L 321: include an 'and' after comma.

L. 348ff: Place these data into a table.

Author Response

(The authors gave the same response as above.)

Round 2

Reviewer 1 Report

Comments and Suggestions for Authors

Dear Authors,

I have carefully reviewed the revised version of your manuscript “Effects of Dietary Supplementation with Dihydromyricetin on the Hindgut Microbiota and Metabolite Profiles in Dairy Cows”. You have addressed all the previous comments thoroughly and appropriately, leading to notable improvements in methodological clarity, interpretative precision, and overall coherence.

The methodological limitations highlighted in the earlier review round have been suitably acknowledged and justified, and they do not compromise the validity of your findings or the strength of your main conclusions.

In light of these revisions, I consider that the manuscript now meets the requirements for publication, and I am pleased to recommend its acceptance.

Yours sincerely,

Author Response

Dear reviewer,

        Thank you so much for handling the review of our revised manuscript (ID microorganisms-4024708) entitled “Effects of Dietary Supplementation with Dihydromyricetin on the Hindgut Microbiota and Metabolite profiles in Dairy Cows”. We sincerely appreciate the time and efforts you have devoted to guiding us in revising and improving our manuscript. 

      Thank you very much for recommending the acceptance of our manuscript. Please do not hesitate to contact us if you have any questions.

With best regards.

Reviewer 2 Report

Comments and Suggestions for Authors

The authors have improved their manuscript! Still, there are a significant issues.

Major:

While coloration is much better, it isn't true that the authors have sufficiently enlarged the fonts in their figures. This must be improved.

The discussion wasn't significantly shortened to maximally 3 pages as requested.

Table 3: What is the advantage of down-regulating protective factors like conjugated linoleic acid and spermine? This wasn't discussed at all.

Any self-critizism and discussion concerning potentially critical effects of DMY (decrease  of conjugated linoleic acids and of spermine) is missing.

It's mostly a high-lighting of preliminary results which  - due to the absence of correction by multiple testing - may be by chance.

Fig. 5: What is the specific information here?

Minor:

L. 122: Table, not Tabel.

L. 124: displays.

L. 126: location and country of company missing.

Fig. 1: The sequence of species indicated should be the same as in the figure, not in opposite direction. So elsewhere.

L. 484: implies, not imply.

L. 535f: grammar/wording of sentence.

Author Response

We sincerely appreciate the time and efforts you have devoted to guiding us in revising and improving our manuscript. Please see the attachment.
